# An Efficient Structural Pruning for Spiking Neural Networks by Balancing Accuracy and Sparsification

## Abstract

The increasing scale of spiking neural networks (SNNs) poses significant challenges for deployment on resource-constrained neuromorphic hardware, necessitating lightweight and learnable structural solutions. Interestingly, biological neural systems employ an efficient organizational strategy—hierarchical structural reorganization around functional clusters, where new connections grow orthogonally to existing ones to expand representational capacity. Inspired by this mechanism, we propose a dynamic pruning and regrowth framework with channel-level orthogonality for SNNs (DPRC-SNNs) to enable scalable and efficient structural learning for SNNs. DPRC-SNNs introduce the spiking column subset selection mechanism for SNNs, which integrates channel-level pruning with orthogonality-driven regrowth, selectively restoring diverse and complementary channels to minimize information loss from aggressive pruning. Through iteratively pruning redundant channels and regrowing orthogonal ones, DPRC-SNNs preserve functional diversity while enhancing sparsity at the channel level. Extensive evaluations on CIFAR10, DVS-Gesture, and DVS-CIFAR10 demonstrate that DPRC-SNNs achieve high compression rates and computational efficiency without compromising accuracy, showing strong potential for neuromorphic deployment.

## 1 Introduction

Spiking neural networks (SNNs) have demonstrated significant potential in replicating the biological efficiency and event-driven processing capabilities of the human brain. Unlike conventional artificial neural networks (ANNs), which rely on continuous activations, SNNs transmit information using discrete spike events to update neuronal membrane potentials over time. This approach makes SNNs particularly well-suited for energy-efficient neuromorphic hardware (Maass, 1997; Indiveri et al., 2011; Davies et al., 2018). As SNNs achieve higher performance, their architectures have become deeper and more complex, with increasing numbers of parameters to address large-scale benchmarks. However, the added depth and complexity result in greater computational demands and memory usage (Roy et al., 2019; Zhu et al., 2022). This trend stands in contrast to biological systems, where neurons and synapses operate under strict resource constraints. In the brain, synaptic connections are limited and continuously pruned and reorganized to maintain efficiency and adapt to evolving requirements. Considering the hardware limitations of neuromorphic chips, specifically the finite number of neurons and synapses available (Indiveri et al., 2011; Davies et al., 2018), there is an increasing need to optimize SNN architectures. Such optimization seeks to reduce network size and power consumption, thereby enhancing the suitability of SNNs for edge computing applications with limited computational resources.

Recent studies have explored structural learning for SNNs, particularly focusing on weight-level optimization. These approaches have demonstrated that SNNs can achieve competitive accuracy with significantly fewer parameters. For instance, inspired by the synapse rewiring mechanism in the brain, (Chen et al., 2021) proposed a method to jointly optimize both the network structure and weights by redefining gradients to manage connectivity. Similarly, (Shen et al., 2023) introduced sparse structural learning for SNNs by employing evolutionary strategies that combine pruning and regrowth of synapses based on the momentum and magnitudes of synaptic connections. Although these weight-level structural learning techniques result in sparse networks, they often require specialized hardware to efficiently support the sparsity of the SNNs. In response to these hardware limitations, recent attention has shifted towards channel-level structural learning for SNNs, which offers more hardware-friendly properties. (Li et al., 2024a) proposes a method that iteratively prunes and regrows channels, achieving sparse yet accurate SNN architectures. Additionally, (Li et al., 2024b) focuses on removing redundancies and regenerating specific convolutional kernels based on spiking activity levels. Despite these advancements, existing channel-level structural learning methods often overlook the fea-

ture representation relationships among different channels, which could potentially limit their effectiveness in further enhancing model performance.

In biological neural systems, structural reorganization does not occur randomly at isolated synapses but is hierarchically organized around functional clusters, which can be regarded as relatively independent feature-processing channels (Fu et al., 2012; Poirazi & Mel, 2001; Houweling & Brecht, 2008). A key mechanism in this process is the orthogonalized growth of new functional clusters—new connections tend to form adjacent to existing strong connections yet remain functionally independent, effectively expanding the neural representation space along new orthogonal dimensions, thereby enhancing network efficiency and capacity (Poirazi & Mel, 2001; Fiete et al., 2004). Inspired by this mechanism, we propose a data-driven channel-level structural learning method for SNNs. Mimicking the biological process, our approach prunes redundant channels and then introduces new ones following an orthogonality principle, thereby compressing network scale while expanding its representational power. This strategy not only maintains performance but also significantly improves energy efficiency, offering a new pathway for deploying high-efficiency SNN models on neuromorphic hardware and edge computing platforms.

In this work, we propose a dynamic channel pruning and regrowth framework for SNNs, inspired by the adaptive structural reorganization of functional clusters in biological neural circuits. As illustrated in Figure 1, the proposed framework achieves network efficiency at the channel granularity, analogous to the selective activation and reorganization of synaptic connection clusters in the brain. The method operates at the channel level, enabling coarse-grained structural learning that is more hardware-friendly and efficient than unstructured sparsity approaches. The main contributions of this paper are summarized as follows:

- The dynamic structural learning framework, the channel level is proposed for SNNs, which employs dynamic pruning and regrowth with channel-orthogonality based on spatiotemporal patterns in SNNs.
- The framework introduces the spiking column subset selection mechanism for SNNs, which integrates channel-level pruning with orthogonality-driven regrowth, selectively restoring diverse and complementary channels to minimize information loss from aggressive pruning.
- Extensive experiments on CIFAR10, DVS-Gesture and DVS-CIFAR10 demonstrate that DPRC-SNNs achieve significant efficiency gains in both storage and computation while maintaining competitive accuracy. Moreover, the channel-level structured sparsity enhances hardware efficiency and facilitates flexible deployment.

## 2 RELATED WORK

**Structure Learning in ANNs.** In recent years, structured pruning has become an effective strategy for compressing ANNs by removing entire components such as filters, channels, or layers, rather than pruning individual weights (Cheng et al., 2024; Ling et al., 2024). This technique enhances computational efficiency and reduces memory usage while maintaining high model accuracy. A key advantage of structured pruning over unstructured pruning lies in its ability to work efficiently with hardware optimizations, as it avoids sparse matrices and takes full advantage of parallel processing capabilities. Recent advances in structured pruning have introduced several powerful methods, such as Gradual Pruning (He et al., 2022), which gradually prunes filters in convolutional networks, ensuring minimal performance loss while achieving significant compression. Another notable method is Group Lasso Pruning (Hoefler et al., 2021), which leverages group sparsity to simultaneously prune entire groups of weights, leading to more structured, efficient models. Moreover, channel pruning (Li et al., 2022) evaluates the importance of individual channels and removes those with minimal contribution to the network's performance, yielding faster and more memory-efficient models. Filter Pruning (He et al., 2022) takes a similar approach but focuses on pruning entire filters within convolutional layers, thereby enhancing computational performance. Recent innovations in automated pruning strategies, such as AutoPrune (Fan et al., 2022), combine reinforcement learning with neural architecture search to autonomously discover optimal pruning strategies, achieving high compression with minimal performance degradation. These advancements represent a significant shift towards more efficient and automated pruning techniques in the field of ANNs model optimization (Hou et al., 2025).

**Structure Learning in SNN.** SNNs offer distinct advantages regarding low energy consumption due to their event-driven nature and sparse temporal activations. Combined with the sparse structure learning methods, SNNs have the potential to implement energy-efficient computing. (Han et al., 2025) introduces a method inspired by biological dendritic spine plasticity, combining neuronal pruning, synaptic constraint, and regeneration to compress SNNs without severely damaging accuracy. (Chen et al., 2021) proposes a training-time method that jointly learns structure and weights by redefining gradients to manage connectivity and enable both pruning and regrowth during training. (Chen et al., 2023) defines a neuron criticality metric inspired by the "critical brain hypothesis" and uses

Figure 1: This schematic illustrates our DPRC-SNNs method, which simultaneously optimizes the weights and explores the sub-model structure within a single training process from scratch. In DPRC-SNNs, both the preserved channels and the regrown channels remain active, jointly participating in the training iterations.

it to guide both structured and unstructured pruning, with regeneration mechanisms to maintain performance even under large pruning ratios. (Li et al., 2024b) constructs a structured pruning framework based on convolutional kernel activity levels; during training, kernels with low activity are pruned, and structure is refined with periodic regeneration to adapt channel counts within layers. (Lew et al., 2023) rocessor design proposes pruning neurons based on their temporal behavior (e.g., membrane voltage thresholds over time), effectively skipping computation for less important neurons in later time steps; this leads to structured neuron pruning that aligns well with temporal redundancy in SNNs. (Li et al., 2024a), which iteratively prunes and regrows channels to obtain sparse yet accurate SNN architectures, achieving significant parameter reduction with minimal accuracy loss. Similarly, (Shen et al., 2023) introduces Evolutionary Structure Learning for SNNs (ESL-SNNs), a dynamic strategy that prunes and regrows synaptic connections during training, enabling the model to learn highly sparse structures from scratch while maintaining competitive performance. These methods demonstrate that structured sparsity can be effectively incorporated into SNNs without relying on large pretrained models, aligning with the goal of efficient and scalable spiking networks.

**Unstructure Learning in SNN.** Existing pruning approaches for spiking neural networks (SNNs) are predominantly *unstructured*, operating at the individual synapse or weight level. (Chen et al., 2021) introduce Gradient Rewiring, which dynamically removes and regrows single connections based on synaptic gradients. (Shi et al., 2024) further explore energy-oriented synaptic sparsity by jointly pruning weights and neurons to reduce firing activity. More recently, (Shi et al., 2025) propose OSBC, a one-shot post–training compression scheme that prunes and quantizes weights based on membrane-potential sensitivity. While these methods achieve high parameter sparsity, their irregular fine-grained patterns incur indexing overhead, offer limited acceleration on general-purpose hardware, and do not explicitly capture the spatiotemporal activation dynamics characteristic of SNNs. In contrast, our work adopts a *structured, channel-level* pruning strategy. By estimating channel importance from spatiotemporal sensitivity and restoring complementary feature channels via orthogonality-driven regrowth, the proposed framework produces hardware-friendly sparsity while preserving temporal representation diversity. This structured formulation directly reduces tensor dimensions, thereby lowering SynOps and memory footprint, improving deployment efficiency, and avoiding the instability commonly observed in highly unstructured SNN pruning.

## 3 PRELIMINARY

### 3.1 SPIKING NEURAL NETWORK

The event-driven computation in SNNs not only makes SNNs biologically plausible but also provides the potential for energy-efficient processing on neuromorphic hardware. A core component of SNNs is the spiking neuron model, which governs how membrane potentials evolve and when spikes are emitted. The Leaky Integrate-and-Fire (LIF)

model is one of the most widely used (Wu et al., 2018);(Xiao et al., 2022) due to its simplicity and biological plausibility. The dynamics of the membrane potential $u(t)$ of a LIF neuron can be described by the following differential equation:

$$\tau_m \frac{du(t)}{dt} = -u(t) + RI(t), \tag{1}$$

where $\tau_m$ is the membrane time constant, $R$ is the membrane resistance, and $I(t)$ denotes the synaptic input current at time $t$. Intuitively, the membrane potential integrates incoming currents and simultaneously leaks over time, mimicking the biophysics of biological neurons.

A spike is emitted whenever the membrane potential crosses a threshold $V_{th}$:

$$s(t) = H(u(t) - V_{th}), \tag{2}$$

where $H(\cdot)$ is the Heaviside step function. After firing, the neuron undergoes a reset process:

$$u(t) \leftarrow u_{reset}, \quad \text{if } u(t) \geq V_{th}, \tag{3}$$

where $u_{reset}$ is often set to zero or a small constant.

For computational implementations, it is common to use a discrete-time approximation of the LIF dynamics, especially in neuromorphic simulations or GPU-based training:

$$u_{t+1} = \alpha u_t + RI_t - V_{th} \cdot s_t, \tag{4}$$

where $\alpha = \exp(-\Delta t/\tau_m)$ is the decay factor controlling the leak, and $s_t$ represents the spike at time step $t$. This formulation explicitly separates the integration, leakage, and reset mechanisms.

The binary and non-differentiable nature of $s_t$ poses challenges for training SNNs with gradient-based methods. To address this, surrogate gradient techniques are widely used, where the derivative of the step function is replaced with a smooth approximation, thus enabling end-to-end optimization of deep SNNs. These dynamics form the foundation for building more complex SNN architectures and for applying advanced optimization methods for structure learning.

## 4 METHODS

Unlike traditional channel pruning methods(Chowdhury et al., 2021);(Lew et al., 2023);(Nguyen et al., 2021), we dynamically adjust channel strength through a periodic pruning and regrowth process, so that channels that were pruned early can be restored and the loss of early representation ability during model retraining can be avoided.

### 4.1 CHANNEL PRUNING STAGE

In SNNs, channel pruning can be framed as a temporal column subset selection problem (Gu & Eisenstat, 1996). Unlike traditional CNNs, where convolutional features are processed in a single pass, SNNs propagate spike-based activations across discrete time steps. Given a convolutional layer in an SNN with weight matrix $\mathbf{W}^l \in \mathbb{R}^{K_l \times C_l}$, where $K_l$ represents the kernel size and $C_l$ is the number of output channels, the feature maps at time step $t$ are given by:

$$\mathbf{Y}_t^l = \mathbf{W}^l * \mathbf{X}_t^{l-1}, \quad t = 1, \ldots, T, \tag{5}$$

where $\mathbf{X}_t^{l-1} \in \mathbb{R}^{C_{l-1} \times H \times W}$ is the input spike tensor at time $t$. For channel pruning, the goal is to select the most representative subset of channels that capture the spatiotemporal dynamics of the input spikes. Formally, we define the desired sparsity for the $l$-th layer as $\kappa_l$, and aim to retain the most informative channels. The pruned channels are selected based on their temporal contribution to the layer's activations.

We introduce a **Spiking Column Subset Selection (SCSS)** approach for pruning in SNNs. SCSS aims to select a subset of columns (channels) from the weight matrix $\mathbf{W}^l$ that best preserves the spatiotemporal information across all time steps. The objective is to minimize the Frobenius norm of the reconstruction error:

$$\mathbf{W}_c^l = \arg\min_{\mathbf{W}_c^l} \sum_{t=1}^T \|\mathbf{W}^l - \mathbf{W}_c^l (\mathbf{W}_c^l)^\dagger \mathbf{W}^l\|_F^2, \tag{6}$$

where $(\cdot)^\dagger$ denotes the Moore-Penrose pseudo-inverse. This approach accounts for the temporal aspect of SNNs by considering the reconstruction error over all time steps, ensuring that channels with strong temporal activations are prioritized for retention.

Next, we compute the leverage scores to quantify the importance of each channel. In SNNs, the weight matrix $\mathbf{W}^l$ operates over $T$ discrete time steps. Thus, the importance of each channel is evaluated not only based on its spatial contribution but also across the temporal domain. To account for the temporal behavior, we compute the leverage score for the $j$-th channel by summing its contribution at each time step, where $\mathbf{U}_t^l$ is the matrix of singular vectors for the $l$-th layer at time step $t$. The leverage score for the $j$-th channel is given by:

$$\ell_j^l = \sum_{t=1}^{T} \left\| [\mathbf{U}_t^l]_{j,:} \right\|_2^2 \tag{7}$$

where $[\mathbf{U}_t^l]_{j,:}$ represents the $j$-th row of the singular vector matrix $\mathbf{U}_t^l$ at time step $t$, and $\| \cdot \|_2$ denotes the L2 norm. This formula sums the contribution of each channel across all time steps, capturing its temporal importance in the context of spiking activity, which is crucial for pruning in SNNs.

In the pruning process, we retain the channels with the highest leverage scores, ensuring that the SNN preserves the most informative spatiotemporal features. This selective pruning approach not only reduces the number of parameters but also maintains the critical temporal dynamics of the network, optimizing both performance and computational efficiency in spike-based processing. For more details on the SCSS formula, please see the Appendix B.

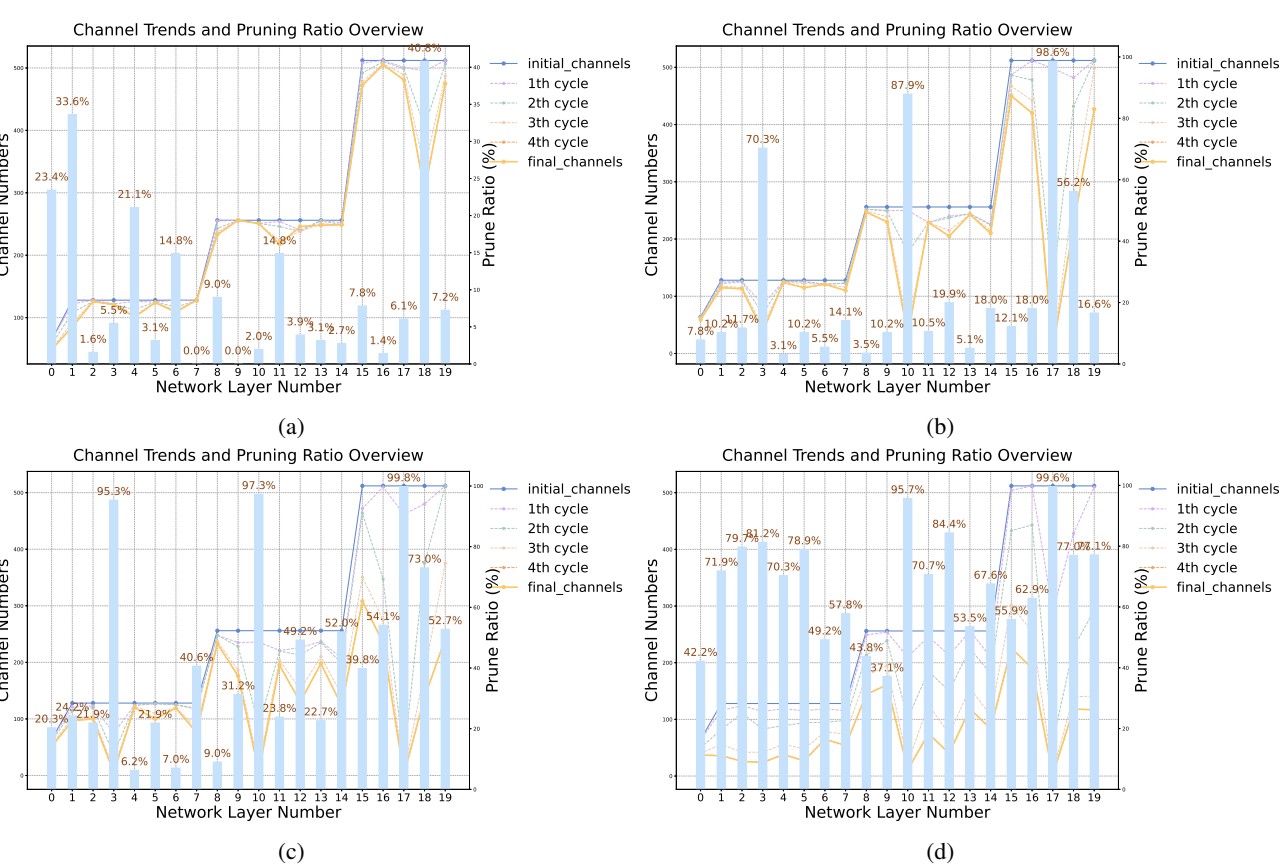

Figure 2: Illustration of pruning settings under different pruning ratios. Subfigure (a), (b), (c), and (d) corresponds to a pruning ratio of 0.1,(b) 0.3, (c) 0.5, and (d) 0.7, showing the layer-wise changes across different training epochs.

## 4.2 CHANNEL REGROWING STAGE

**Channel Regrowth in SNNs.** To mitigate the sub-optimality of early pruning decisions in SNNs, we introduce a regrowth mechanism that periodically reactivates a subset of previously pruned channels. Let $W_l \in \mathbb{R}^{K \times C_l}$ denote the weight matrix of layer $l$ and $M_l \in \{0, 1\}^{C_l}$ the corresponding channel mask. After pruning, a subset of channels is deactivated, and for each pruned channel $j$ we store a snapshot $\widehat{W}_{l,j}$ of its last active state.

A critical step in regrowth is how to assign weights to the reactivated channels. A naive solution is zero initialization, which ensures no immediate change to the network's output. However, in SNNs such channels rarely fire due to their membrane potentials staying below threshold, and thus they receive vanishing surrogate gradients in subsequent training. This makes them unable to recover effectively.

To address this issue, we restore the most recently used parameters of the pruned channels:

$$W_{l,j}^{(t+1)} \leftarrow \widehat{W}_{l,j}, \qquad j \in \mathcal{R}_l, \tag{8}$$

where $\mathcal{R}_l$ denotes the set of reactivated channels. By resuming from their last informative state, the regrown channels can actively contribute to spatiotemporal feature encoding across time steps and be properly evaluated in subsequent pruning stages.

Moreover, to progressively stabilize the pruning–regrowth process, we employ a cosine-decayed regrowth factor that gradually reduces the number of reactivated channels as training proceeds. At the $t$-th pruning step, the regrowth factor is given by:

$$\delta_t = \delta_0 \cdot \frac{1}{2} \left( 1 + \cos \left( \frac{\pi t}{T_{\max}/\Delta T} \right) \right), \tag{9}$$

where $\delta_0$ is the initial regrowth budget, $T_{\max}$ denotes the total exploration steps, and $\Delta T$ controls the frequency of pruning–regrowth cycles. This schedule ensures that the sub-model gradually converges toward the target channel sparsity, while still allowing sufficient exploration in the early training stages.

Table 1: The performance comparison between DPRC-SNNs and other SNNs models

| Dataset | Method | Architecture Network | Acc (%) | Acc Loss(%) | Connection Density(%) |
|---|---|---|---|---|---|
| CIFAR10 | ADMM-base (Deng et al., 2021) | 7Conv+2FC | 89.53 | -3.85 | 10 |
| | Grad R (Chen et al., 2021) | 6Conv+2FC | 92.84 | -0.34 | 12 |
| | TET[1] (Deng et al., 2022) | ResNet-19 | 92.79 | - | - |
| | ESL-SNN (Shen et al., 2023) | Sparse-ResNet19 | 91.09 | -1.70 | 50.00 |
| | SCA-based (Li et al., 2024b) | VGG16 | 91.14 | -0.88 | 9.31 |
| | Neuron Pruning (Li et al., 2024a) | Resnet18 | 92.91 | -0.01 | 89.36 |
| | Channel Pruning (Li et al., 2024a) | VGG16 | 91.24 | -0.47 | 77.17 |
| | PQ-SNN (Shen et al., 2025) | ResNet19 | 92.38 | +0.11 | 29.72 |
| | **DPRC-SNNs** | ResNet19-SNN | **93.29** **92.64** | **+0.24** **-0.41** | **70** **50** |
| DVS-Gesturte | Neuron Pruning (Li et al., 2024a) | VGG13 | 94.44 | - | 50 |
| | Grad R (Chen et al., 2021) | 2Conv+2FC | 84.12 | 0.00 | 50.00 |
| | **DPRC-SNNs** | Resnet19-SNN | **96.88** | **+1.05** | 49.80 |
| DVS-CIFAR10 | ELS-SNN (Shen et al., 2023) | VGG8 | 78.3 | -0.28 | 10 |
| | SCA-based (Li et al., 2024b) | 5Conv+1FC | 72.8 | +0.9 | 21.73 |
| | TET (Deng et al., 2022) | VGGSNN | 83.17 | - | - |
| | PQ-SNN (Shen et al., 2025) | VGGSNN | 78.4 | -1.4 | 4.46 |
| | **DPRC-SNNs** | ResNet19-SNN | **81.50** **82.10** | -0.80 -0.20 | **50** **70** |

We compare our pruning method with TET because it is a strong and widely adopted training paradigm for improving SNN accuracy. Using TET as the baseline ensures a fair comparison.

**Channel Regrowth via Orthogonality.** In previous studies on SNNS pruning with regeneration mechanisms (Han et al., 2025);(Han et al., 2024), the regrowth of pruned synaptic connections was often implemented through simple activity-based heuristics or a uniform random sampling of candidate connections. While such strategies can partially

restore network capacity, they inherently suffer from two major drawbacks. First, they do not explicitly consider the redundancy among regrown channels, which may lead to reintroducing connections that are highly correlated with the already preserved ones. This results in a limited contribution to improving the diversity of feature representations. Second, in the temporal domain of SNNs, such naive regrowth fails to guarantee the recovery of cross-time-step feature propagation, thereby risking the loss of important temporal dynamics.

To address these limitations, we propose a regrowth mechanism based on *orthogonal projection*, which is inspired by the biological principle of synaptic competition and decorrelation in cortical circuits. In order to incorporate the temporal dynamics of SNNs, we compute the orthogonality score of a candidate channel $w_j^l$ with respect to the active channel subspace $W_T^l$ over $T$ discrete time steps as

$$\Omega_j^l = \frac{1}{T} \sum_{t=1}^{T} \left\| \left( I - W_T^l \left( (W_T^l)^\top W_T^l \right)^\dagger (W_T^l)^\top \right) \left( w_j^l \cdot X^{l-1}(t) \right) \right\|_2^2, \tag{10}$$

where $X^{l-1}(t)$ denotes the input spikes at time step $t$, $\cdot$ is the convolution or linear transformation, and $\dagger$ represents the Moore–Penrose pseudoinverse. This formulation measures the novelty of the candidate channel in the temporal dimension of the SNN.

Next, we define an *importance sampling distribution* over the pruned channels based on the orthogonality scores:

$$p_j^l = \frac{\exp(\Omega_j^l)}{\sum_{j' \in [C_l] \setminus T_l} \exp(\Omega_{j'}^l)}, \quad j \in [C_l] \setminus T_l. \tag{11}$$

The set of channels to regrow is then sampled without replacement according to a multinomial distribution:

$$\mathcal{R}^l \sim \text{Multinomial}\left( \{p_j^l\}_{j \in [C_l] \setminus T_l}; \lfloor \delta_t C_l \rfloor \right), \tag{12}$$

where $\delta_t$ is the regrowth factor at iteration $t$, controlling the fraction of channels to be reactivated.

## 4.3 DYNAMIC CHANNEL REGROWTH AND STRUCTURE EXPLORATION

The initial architecture of SNNs may not exhibit a balanced channel distribution across layers. Some layers contribute more critically to the spatiotemporal feature representation, while others contain redundant channels. To preserve accuracy under pruning, we perform **dynamic sub-model structure exploration**, which reallocates surviving channels across layers based on both batch normalization (BN) scaling factors and spiking activity, where the temporal nature of the spikes is leveraged.

Specifically, we define a *spike-aware importance score* for each layer, which captures the spiking activity in conjunction with the static scaling factors from BN:

$$\phi_l = \|\gamma_l\|_1 \cdot \rho_l, \tag{13}$$

where $\gamma_l$ denotes the BN scaling factors of layer $l$ (Liu et al., 2017), and $\rho_l$ represents the average spike firing rate of the layer, which is computed based on the temporal spike activity:

$$\rho_l = \frac{1}{T} \sum_{t=1}^{T} \frac{1}{H_l W_l} \sum_{i,j} \mathbf{Y}_{t,i,j}^l, \tag{14}$$

where $T$ is the number of time steps, $H_l$ and $W_l$ are the height and width of the feature map, and $\mathbf{Y}_{t,i,j}^l$ denotes the spike output of the $i$-th and $j$-th neurons at time step $t$ in layer $l$.

This formulation integrates the static scaling information from BN with the dynamic temporal activity of spiking neurons, ensuring that layers with both strong scaling responses and rich spiking dynamics are prioritized for retaining channels.

Given an overall target sparsity $S$, the layer-wise pruning ratio $\kappa_l$ is then computed as:

$$\kappa_l = 1 - \frac{\phi_l}{\sum_{j=1}^{L} \phi_j} \cdot (1 - S), \tag{15}$$

where $L$ is the total number of layers. Intuitively, layers with higher $\phi_l$ values, which reflect both strong spiking activity and significant scaling responses, retain a larger portion of their channels, while less important layers are pruned more aggressively.

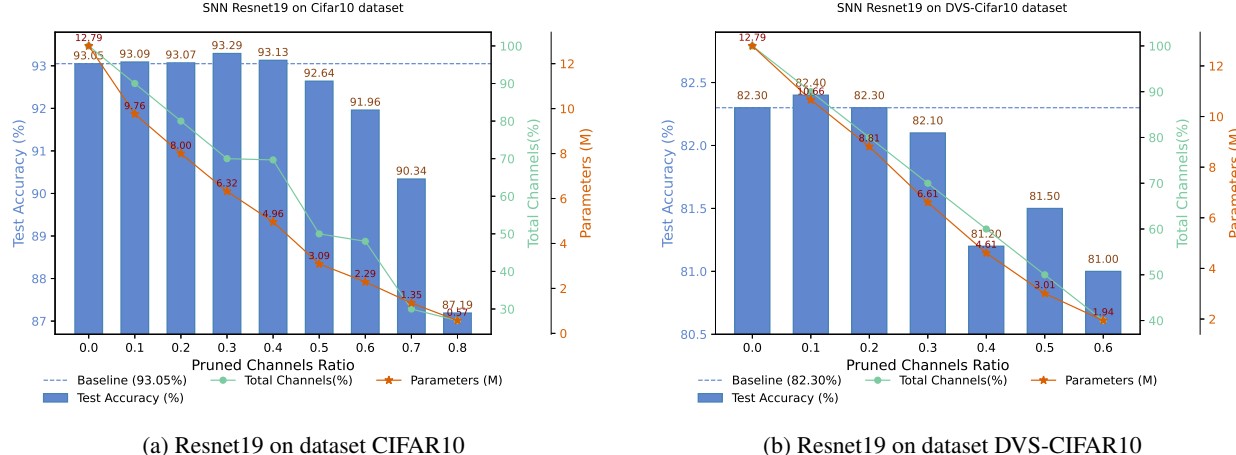

(a) Resnet19 on dataset CIFAR10            (b) Resnet19 on dataset DVS-CIFAR10

Figure 3: The performance of the DPRC-SNNs structure learning framework.

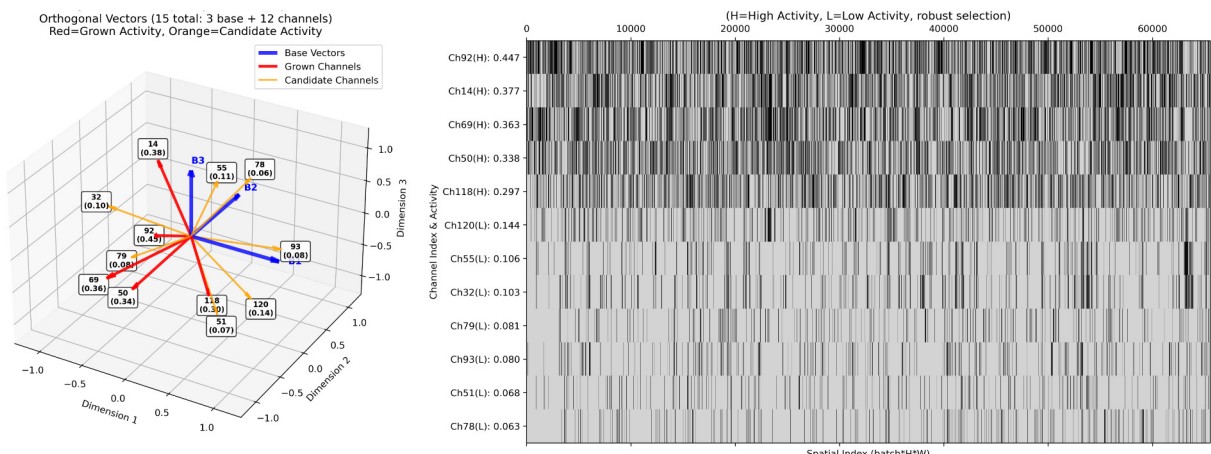

Figure 4: The spike intensity emitted by the orthogonal growth channel and candidate channel of the DPRC-SNNs during the regrowth process is represented.

During training, this reallocation is performed iteratively in tandem with regrowth. The regrowth stage enlarges the candidate set of channels, while the exploration stage dynamically redistributes them across layers to adapt to the spatiotemporal nature of the spikes. This synergy allows the model to preserve critical spiking neurons while pruning redundant features, optimizing both temporal and spatial dynamics for efficient SNNs pruning. A detailed stability and convergence analysis of this mechanism, including the smoothness of importance scores, Lipschitz continuity of pruning ratios, and contraction bounds on pruning–regrowth iterations, is provided in Appendix C.

## 5 EXPERIMENTS

We evaluate the proposed DPRC-SNNs algorithm on both static and neuromorphic datasets and compare it with existing methods. CIFAR10 is a widely used benchmark for static image classification, containing 10 classes. All images are $32 \times 32$ RGB images, which need to be encoded before being fed into the SNN. For the neuromorphic benchmark, we use DVS-CIFAR10, splitting it into 9,000 training samples and 1,000 testing samples. We set the initial regrowth factor and the interval as $\Delta T = 20$ training epochs, where the cycle is fixed to 4 throughout the experiments. Here, $\delta_0$ denotes the pruning ratio, while $\Delta T$ represents the number of training iterations between two consecutive pruning–regrowth steps. For simplicity and generality, these hyperparameters are kept constant throughout all experiments.

## 5.1 EFFECTIVENESS ANALYSIS

**Performance Comparison.** Table 1 summarizes the performance of DPRC-SNNs across multiple datasets. Different channel pruning ratios are applied, maintaining a fixed sparsity level during training. After training, a new sparse SNN structure is obtained and evaluated. Model efficiency is assessed in terms of parameter count and spike operations (SOPs), both of which serve as proxies for memory footprint and energy consumption. On CIFAR10, we train for only 200 epochs and successfully compress the model to a 30% connection ratio. Remarkably, this configuration not only achieves a 0.24% accuracy improvement over the full model (baseline accuracy: 93.05%), but also yields an extremely low computational cost of 66.49K SOPs. This is orders of magnitude lower than the SOPs reported by current state-of-the-art structured sparsity methods—e.g., the SCA-based approach (Li et al., 2024b), which requires 90.82K SOPs under comparable accuracy levels. These results highlight the superior computational efficiency of our DPRC-SNNs under spatiotemporal sparsity.

This clearly demonstrates the effectiveness of our method in static image recognition. Furthermore, we validate DPRC-SNNs on the challenging DVS-CIFAR10 dataset, where the model achieves an impressive 82.10% accuracy under 30% connection pruning, outperforming all existing methods to date and once again confirming the superiority of our approach. Additionally, when evaluated on the DVS-Gesture dataset, a highly challenging neuromorphic benchmark, our method achieves an impressive accuracy of 96.88% even under 49.8% pruning, further showcasing the robustness and effectiveness of DPRC-SNNs across diverse neuromorphic datasets.

**Structural Analysis.** To better understand the learning process, we visualize the channel count per layer in ResNet19 across different pruning rounds (Fig 2). Each curve represents the number of channels in a given layer after a pruning step on CIFAR10. The evolution of channel counts across successive pruning rounds reveals clear and consistent trends. In the early pruning stages, the initial layers have already learned strong feature representations, so the pruning ratios of the later layers are relatively higher. As the overall pruning ratio increases, all layers progressively remove redundant channels to meet the target compression. The final layers are pruned more conservatively to preserve high-level semantic features crucial for classification, as excessive pruning here would significantly harm model performance. Layers 3, 7, and 10 experience the most pruning, as they correspond to downsampling stages with a large number of channels. Many of these channels are redundant due to reduced spatial resolution, making them ideal candidates for pruning without significantly impacting the network's representational capacity. As training and pruning progress, the model structure gradually stabilizes, indicating that the structural learning framework adaptively converges to an appropriate architecture over iterative pruning.

To evaluate the model's regrowth capabilities, we conduct orthogonal analysis on selected channels within specific layers, as illustrated in (Fig. 4). The orthogonal projection is computed relative to a basis vector (shown in blue) according to (Eq 10). Channels exhibiting higher orthogonal values demonstrate greater spike intensity and enhanced feature independence, indicating their necessity for regrowth during the pruning process. Conversely, channels with lower orthogonal values show reduced independence and can be safely pruned without regrowth. The spike activation patterns presented in the right panel provide empirical validation of this orthogonality-based channel selection criterion, where high-activity channels (marked in red) clearly demonstrate distinct firing patterns compared to low-activity channels (marked in orange), confirming the effectiveness of our orthogonal analysis for identifying critical channels requiring preservation and regrowth.

## 5.2 ABLATION STUDY

To further validate the effectiveness of DPRC-SNNs in structure learning, we conduct ablation experiments on CIFAR10 and DVS-CIFAR10, as shown in Figure 3. The blue bars represent the test accuracy of SNN Resnet19 on the datasets CIFAR10 and DVS-CIFAR10, while the orange curves show the number of parameters at different pruning ratios. The blue curve indicates the accuracy of the unpruned baseline model, and the green curve depicts the number of channels at different pruning ratios. When the pruning ratio reaches 0.6 on the CIFAR10 dataset, the model accuracy only drops by 1.09%. On DVS-CIFAR10, pruning 20% of the parameters does not affect accuracy, and even at 50% pruning, the model accuracy decreases by just 0.80%, with the parameter count reduced to 1/4 of the previous one. In contrast, pruning-only training shows minimal mask updates after the initial step, highlighting that the regrowth mechanism in DPRC-SNNs is crucial for reactivating incorrectly pruned channels. These results confirm that DPRC-SNNs are robust and stable across a wide range of pruning ratios, making them an effective solution for structured pruning with both high performance and efficiency.

## 6 CONCLUSION

The depth and complexity of SNNs have been expanding across diverse applications, which, in turn, hinders their potential for low energy efficiency due to parameter redundancy and high memory requirements. In this work, we introduce DPRC-SNNs, a novel framework designed for training sparse SNNs at the channel level. DPRC-SNNs implement a spiking column subset selection strategy that integrates channel-level pruning with orthogonality-driven regeneration. This approach selectively reintroduces diverse and complementary channels to minimize the information loss resulting from aggressive pruning. By systematically pruning redundant channels and regenerating orthogonal ones, DPRC-SNNs preserve functional diversity while promoting greater sparsity at the channel level. Experimental results demonstrate that DPRC-SNNs successfully learn compact, sparse architectures that achieve competitive accuracy with significantly fewer parameters. Moreover, sparse training at the channel level enhances the expressive power of the learned network, offering substantial benefits for embedded hardware, including reduced power consumption, lower memory usage, and improved on-chip learning efficiency.

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

## A APPENDIX

### A.1 USE OF LLMs

Large Language Models (LLMs) were used solely to assist with polishing the text.

### A.2 CODE OF ETHICS AND ETHICS STATEMENT

The research conducted in the paper conforms, in every respect, with the ICLR Code of Ethics https://iclr.cc/public/CodeOfEthics.

### A.3 THE SUPPLEMENTARY MATERIALS FOR THE PRELIMINARY OF SNNs

**Surrogate Gradient Training.** As mentioned above, the major difficulty in training SNNs arises from the non-differentiability of the spike generation function $s(t) = H(u(t) - V_{th})$, since the Heaviside step function $H(\cdot)$ has zero gradient almost everywhere. This prevents the direct application of gradient-based optimization methods such as backpropagation.

To address this, surrogate gradient methods approximate the derivative of the spike function with a smooth surrogate during the backward pass, while keeping the exact binary spike function in the forward pass. Formally, let $s(t) = H(u(t) - V_{th})$. In surrogate gradient training, the forward and backward computations are decoupled:

$$\frac{\partial s(t)}{\partial u(t)} \approx \sigma'(u(t) - V_{th}), \tag{16}$$

where $\sigma(\cdot)$ is a smooth function, such as a sigmoid, piecewise linear, or exponential function, used only for gradient computation. For example, a common surrogate is the fast sigmoid derivative:

$$\sigma'(x) = \frac{1}{(1 + \beta|x|)^2}, \tag{17}$$

where $\beta$ controls the slope sharpness. Another popular choice is the piecewise linear approximation:

$$\sigma'(x) = \begin{cases} 1 - |x|/\gamma, & |x| < \gamma, \\ 0, & \text{otherwise,} \end{cases} \tag{18}$$

where $\gamma$ defines the surrogate window.

These approximations allow error signals to propagate through spiking neurons, enabling end-to-end supervised training of deep SNNs on large-scale datasets. This surrogate gradient framework has become the standard approach for modern SNN optimization.

## B   MORE DETAILS ON SCSS MATRIX APPROXIMATION AND SUBSPACE SELECTIONIN

Below, we clarify how our SCSS formulation relates to classical matrix approximation and subspace-selection techniques, which originated from numerical linear algebra and have been widely used in CNN model compression. Low-rank decomposition is commonly applied to compress CNN convolution kernels(Denton et al., 2014). Given a weight matrix:$W \in \mathbb{R}^{K \times C}$,its truncated SVD yields:   $W \approx U_r \Sigma_r V_r^\top$,where $\tau$ is the target rank. Although optimal in Frobenius norm, this method does **not select actual channels** (columns of $w$), making it unsuitable for channel pruning.

The classical column subset selection (Gu & Eisenstat, 1996) seeks a subset of columns $C$ of $W$ that best reconstruct the full matrix:

$$C^\star = \arg \min_C \left\| W - CC^\dagger W \right\|_F^2 \tag{19}$$

CUR decomposition (Mahoney & Drineas, 2009) expresses:$W \approx CUR$ , where $C$ contains a subset of real columns of $W$, directly corresponding to selected CNN channels.

Leverage scores are widely used to approximate column subset selection (Drineas et al., 2006). For truncated SVD $W \approx U_r \Sigma_r V_r^\top$

$$\ell_j = \|V_r(j,:)\|_2^2 \tag{20}$$

This represents the energy of each channel within the dominant subspace. However, CNNs compute such importance only **once**, since their activation is purely spatial and static (single-pass).

Unlike CNNs, SNNs propagate information across $T$ discrete time steps, and the importance of a channel depends

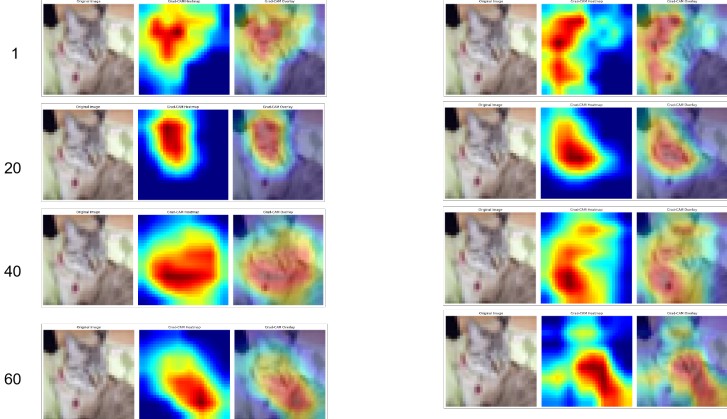

Figure 5: Grad-CAM comparison of pruned (left) and unpruned (right) models. The pruned model focuses on the most discriminative regions, while the unpruned model shows more diffuse, less representative attention.

on its contribution to the spatiotemporal evolution of membrane potentials and spikes. Directly applying CNN-style column subset selection or leverage scores ignores this temporal dynamics. Let $W_l$ be the weight matrix reused at each time step $t = 1, \ldots, T$. We extend classical column subset selection by minimizing the reconstruction error across all time steps:

$$W_{l,c}^\star = \arg \min_{W_{l,c}} \sum_{t=1}^{T} \left\| W_l - W_{l,c}(W_{l,c})^\dagger W_l \right\|_F^2 . \tag{21}$$

This temporal objective is unique to SNNs, since CNNs do not maintain time-varying activations. At each time step, compute the left singular vectors:

$$W_l \approx U_{l,t} \Sigma_{l,t} V_{l,t}^\top. \tag{22}$$

We then define the temporal leverage score for channel $j$ as:

$$\ell_j = \sum_{t=1}^{T} \|U_{l,t}(j,:)\|_2^2. \tag{23}$$

This score measures how consistently a channel contributes to the dominant temporal subspaces, highlighting channels with strong and stable activations across all $T$ steps. CNNs cannot obtain this score because they lack a temporal dimension.

## C  STABILITY AND CONVERGENCE ANALYSIS OF THE ADAPTIVE SPARSITY MECHANISM

The proposed adaptive sparsity mechanism relies on both batch normalization (BN) scaling factors and the temporal spiking activity of SNNs. We show that the interaction of these two signals leads to stable pruning dynamics and does not introduce optimization instability.

Table 2: Testing on more architectures or datasets

| Dataset | Method | Architecture Network | Acc (%) | Acc Loss(%) | Connection Density(%) |
|---|---|---|---|---|---|
| Tiny-Imagenet | Attention-base (Deng et al., 2021) | VGG16 | 51.92 | +0.78 | 40 |
| | SCA-based (Li et al., 2024b) | VGG16 | 49.33 | -0.19 | 30.60 |
| | **DPRC-SNNs** | VGGSNN | **62.33** **60.92** | **-0.01** **-1.42** | **70** **50** |
| CIFAR100 | ELS-SNN (Shen et al., 2023) | Sparse ResNet-19 | 73.48 | -0.99 | 50 |
| | SCA-based (Li et al., 2024b) | VGG16 | 64.89 | +0.64 | 23.52 |
| | ANN | ResNet-19 | 75.35 | - | - |
| | TET (Deng et al., 2022) | ResNet-19 | 74.47 | - | - |
| | **DPRC-SNNs** | ResNet19-SNN | **77.21** **75.12** | **+0.02** -2.07 | **77.9** **58.7** |
| CIFAR10 | Spikeformer (Zhou et al., 2022) | Spikeformer-4-384 | 95.19 | - | - |
| | ANN | Transformer-4-384 | 96.73 | - | - |
| | **DPRC-SNNs** | **Spikeformer-4-384** | **94.98** | **-0.21** | **50** |

Recall the spike-aware importance score Eq. 13 used in the main text, where $\gamma_l$ is the BN scaling vector and $\rho_l$ is the average firing rate defined in Eq. 14. These two terms evolve smoothly during training due to their distinct update characteristics in SNNs. BN scales $\gamma_l$ follow gradient descent with Lipschitz-continuous updates, while the firing rate is a bounded empirical average of spike trains, i.e.,

$$0 \leq \rho_l \leq 1, \qquad |\rho_l^{(k+1)} - \rho_l^{(k)}| \leq C_\rho \eta, \tag{24}$$

where the constant $C_\rho$ depends on the surrogate gradient used in backpropagation through spikes. Because both components vary smoothly, the importance score also changes smoothly:

$$|\phi_l^{(k+1)} - \phi_l^{(k)}| \leq \|\gamma_l\|_1 |\Delta \rho_l| + \rho_l \|\Delta \gamma_l\|_1 = \mathcal{O}(\eta), \tag{25}$$

which means that SNN-specific temporal dynamics do not introduce abrupt jumps in layer importance. The pruning ratio is obtained by normalizing $\{\phi_l\}$:

$$\kappa_l = 1 - \frac{\phi_l}{\sum_{j=1}^{L} \phi_j}. \tag{26}$$

Differentiating with respect to $\phi$ gives bounded Jacobian entries:

$$\left| \frac{\partial \kappa_l}{\partial \phi_m} \right| = \mathcal{O}\left( \frac{1}{(\sum_j \phi_j)^2} \right), \tag{27}$$

implying that the mapping from spike-driven importance scores to pruning ratios is Lipschitz-continuous. Thus, the temporal fluctuations of spikes influence pruning *smoothly*, ensuring that sparsity allocation does not oscillate across training iterations. We further consider the interaction between pruning masks and optimization. For masked SGD updates

$$\theta_{t+1} = \theta_t - \eta(m_t \odot g_t), \tag{28}$$

The stability depends on how quickly the masks change. Since $\kappa_l$ and hence the masks evolve smoothly (due to the bounded updates of $\phi_l$), we have

$$\|m_{t+1} - m_t\|_2 = \mathcal{O}(\eta), \tag{29}$$

which keeps the effective gradient energy nearly unchanged and avoids destructive resets of temporal membrane potentials—an SNN-specific risk during pruning. Finally, over pruning-regrowth cycles indexed by $k$, the sparse model $\mathcal{M}^{(k)}$ satisfies

$$\|\mathcal{M}^{(k+1)} - \mathcal{M}^{(k)}\| = \mathcal{O}(\eta), \tag{30}$$

leading to a contraction-like convergence:

$$\|\mathcal{M}^{(k+1)} - \mathcal{M}^\star\| \leq \alpha\|\mathcal{M}^{(k)} - \mathcal{M}^\star\| + \mathcal{O}(\eta), \quad \alpha < 1. \tag{31}$$

This shows that the adaptive sparsity mechanism—driven jointly by BN scaling and spike dynamics—preserves training stability and converges reliably, while dynamically selecting the most informative temporal-spatial channels in SNNs.

