# OpenReview forum: "An Efficient Structural Pruning for Spiking Neural Networks by Balancing Accuracy and Sparsification"
_ICLR.cc/2026/Conference — Submitted to ICLR 2026_

### Official Review · Reviewer_nFzZ · 2025-10-30

**Soundness:** 2
**Presentation:** 2
**Contribution:** 1
**Rating:** 0
**Confidence:** 4

**Summary:**

This paper presents DPRC-SNNs, a Dynamic Pruning and Regrowth with Channel-level Orthogonality framework for spiking neural networks. Motivated by the hierarchical structural reorganization observed in biological neural circuits, the method enables adaptive structural learning by iteratively pruning redundant channels and regrowing new, orthogonal ones.

**Strengths:**

Structured and hardware-friendly sparsity

**Weaknesses:**

1. The overall pruning–regrowth paradigm has been explored in several works. The main innovation here—orthogonality-driven regrowth—while conceptually appealing, may require stronger empirical justification to establish substantial novelty.
2. This paper does not specify how orthogonality between channels is measured or enforced.
3. This paper should benchmark against modern event-based or structured sparsity approaches.
4. Although the method is motivated by neuromorphic deployment, the results are limited to simulation-level metrics (accuracy, FLOPs). Energy measurements or mapping to actual chips (e.g., Loihi 2, Tianjic) would substantiate the hardware efficiency claim.

**Questions:**

see weakness

---

> ### Author Response · Authors · 2025-11-24
>
> We would like to express our sincere gratitude to the reviewers for their thoughtful and constructive feedback. Below, we provide detailed point-by-point responses to all comments.\
> **W1: About the substantial novelty.**\
> Thanks for the question. The novelty of our method lies in its holistic design for spiking neural networks, which explicitly accounts for the temporal dynamics of spike-based activations by formulating channel selection as a temporal column subset selection problem. To mitigate sub-optimal early pruning, we introduce an orthogonality-driven regrowth mechanism that reactivates channels contributing novel and decorrelated information across time. Combined with a spike-aware importance scoring that integrates batch normalization scaling factors, our approach adaptively reallocates channels across layers, prioritizing those critical for rich spatiotemporal representation. By integrating temporal reconstruction, leverage-based selection, and orthogonal regrowth, our framework provides a temporally-consistent, SNN-specific pruning strategy that optimizes both computational efficiency and the preservation of essential spatiotemporal features.\
> On one hand, most of the existing pruning and regrowth paradigm is designed for the unstructured pruning of SNNs, which are heuristic, typically random, activity-based, or uniformly sampled (Shi et al., 2024; 2025). These existing methods ignore both channel redundancy and temporal spike dynamics, which thus reintroduce highly correlated channels during growth and fail to recover cross-time-step information flow. Differently, our method introduces orthogonality-driven regrowth based on a temporal subspace projection (Eq. 10):$\Omega^l_j \= \frac{1}{T} \sum_{t=1}^{T}
> \left\| \left( I - W^l_T \left( (W^l_T)^\top W^l_T \right)^{\dagger} (W^l_T)^\top \right) \big( w^l_j \cdot X^{l-1}(t) \big) \right\|_2^2$, which selects channels that contribute new spatiotemporal representations.
> On the other hand, the proposed temporal-orthogonal mechanism has no counterpart in ANN pruning, which is designed for SNNs by considering the temporal dynamics. We restore pruned channels using their last active state (Eq. 8):$W^{(t+1)}\_{l,j}\leftarrow \widehat{W}\_{l,j},\qquad j\in\mathcal{R}\_l$, preventing the vanishing-gradient problem caused by silent spikes—an issue fundamentally unique to SNNs and absent in ANNs. \
> Therefore, although the high-level concept of pruning-regrowth exists, our approach is substantially novel because it is explicitly designed for the structural learning of SNNs by considering the temporal spiking dynamics and orthogonality-driven updating.\
> [1] Shi, L., Li, A., & Ward-Cherrier, B. (2025). Optimal Spiking Brain Compression: Improving One-Shot Post-Training Pruning and Quantization for Spiking Neural Networks. arXiv preprint arXiv:2506.03996.\
> [2] Shi, X., Ding, J., Hao, Z., & Yu, Z. (2024). Towards energy-efficient spiking neural networks: An unstructured pruning framework. In The Twelfth International Conference on Learning Representations.\
> **W2:How to measure the orthogonality between channels.**\
> Thanks for the question. We explicitly define how orthogonality between channels is measured in Line 309 Page 6. The orthogonality score is formally given in Eq. (10), where we compute the novelty of a candidate regrowth channel by projecting it onto the temporal subspace spanned by the currently active channels. Specifically, we use the projection operator $\left( I - W^l_T \left( (W^l_T)^\top W^l_T \right)^{\dagger} (W^l_T)^\top \right)$, which measures the component of a channel that is orthogonal to all existing active channels across T discrete timesteps. The resulting squared projection norm serves as the orthogonality score. This mechanism does not enforce orthogonality through a loss term, but rather selects channels whose temporal representations lie outside the current subspace, ensuring maximal diversity during regrowth. By sampling channels according to this orthogonality-based
> importance distribution (Eq. 11):$p^l\_j \= \exp(\Omega^l\_j) \bigg/ \sum\_{j' \in [C\_l] \setminus T\_l} \exp(\Omega^l\_{j'}),
> \qquad j \in [C\_l] \setminus T\_l$, our method effectively encourages decorrelated feature reactivation without requiring explicit constraints or additional regularization.

---

> > ### Author Response · Authors · 2025-11-24
> >
> > **W3:Benchmark against modern even-based or structured sparsity approaches.**\
> > Thank you for pointing out that. Actually, we have already compared DPRC-SNNs against a comprehensive set of modern structured sparsity methods (in Table1).
> > Many structured pruning methods in ANNs (e.g., ResRep (Ding et al., 2020), OICSR (Li et al., 2019), and Efficient Inference via Channel Pruning (Zhang et al., 2019)) reduce FLOPs and parameter counts by statically removing channels or neurons. In contrast, structured pruning in SNNs, as in DPRC-SNNs, must account for spike-based computation and temporal dynamics. Our method not only prunes redundant channels and neurons but also dynamically regrows them based on orthogonality, which preserves cross-time-step information flow while reducing SynOps. This dynamic adaptation makes DPRC-SNNs more energy-efficient than static ANN pruning in spike-based deployment scenarios, while maintaining high accuracy. Thus, DPRC-SNNs combines the hardware-friendly benefits of structured pruning with temporal flexibility unique to SNNs.\
> > In the SNN pruning section, we also compared our method with structured and unstructured pruning in SNNs, as shown in the table below.
> > | Dataset | Type         | Method           | Acc(%) | Connection(%) | Sops      |
> > |---------|--------------|----------------|--------|---------------|-----------|
> > | CIFAR10 | Unstructured | GradR[4]        | 92.03  | 11.85         | 143.69M   |
> > |         |              | weight-Pruning[5]| 92.05  | 1.16          | 9.56M     |
> > |         |              | ADMM[6]         | 90.19  | 25.03         | 107.97M   |
> > | CIFAR10 | Structured   | SCA-base[7]     | 91.14  | 9.31          | 90.82K    |
> > |         |              | ESL-SNN[8]      | 92.08  | 32.71         | 2.53M     |
> > |         |              | PQ-SNN[9]       | 92.38  | 29.72         | 110.65M   |
> > |         |              | DPRC-SNNs       | 92.64  | 50.00         | 51.19K    |
> >
> > | Dataset      | Type         | Method         | Acc(%) | Connection(%) | Sops      |
> > |-------------|--------------|----------------|--------|---------------|-----------|
> > | DVS-CIFAR10 | Unstructured | STDS           | 81.10  | 10.17         | 76.56M    |
> > |             |              | weight-Pruning | 81.00  | 4.68          | 31.86M    |
> > | DVS-CIFAR10 | Structured   | ESL-SNN        | 80.60  | 36.19         | 38.32M    |
> > |             |              | PQ-SNN         | 78.40  | 4.46          | 189.02M   |
> > |             |              | DPRC-SNNs      | 81.05  | 50.00         | 1.23M     |
> >
> > DPRC-SNNs achieve superior or comparable accuracy–sparsity trade-offs across different kinds of datasets, including static images and event-based datasets. \
> > [1]Ding, X., Hao, T., Tan, J., Liu, J., Han, J., Guo, Y., & Ding, G. (2021). Resrep: Lossless cnn pruning via decoupling remembering and forgetting. In Proceedings of the IEEE/CVF international conference on computer vision (pp. 4510-4520).\
> > [2]Li, J., Qi, Q., Wang, J., Ge, C., Li, Y., Yue, Z., & Sun, H. (2019). OICSR: Out-in-channel sparsity regularization for compact deep neural networks. In Proceedings of the IEEE/CVF conference on computer vision and pattern recognition (pp. 7046-7055).\
> > [3]Zhang, B., Davoodi, A., & Hu, Y. H. (2019). Efficient inference of CNNs via channel pruning. arXiv preprint arXiv:1908.03266.\
> > [4]Chen, Y., Yu, Z., Fang, W., Huang, T., & Tian, Y. (2021). Pruning of deep spiking neural networks through gradient rewiring. arXiv preprint arXiv:2105.04916.\
> > [5]Shi, X., Ding, J., Hao, Z., & Yu, Z. (2024). Towards energy efficient spiking neural networks: An unstructured pruning framework. In The Twelfth International Conference on Learning Representations.\
> > [6]Deng, L., Wu, Y., Hu, Y., Liang, L., Li, G., Hu, X., ... & Xie, Y. (2021). Comprehensive snn compression using admm optimization and activity regularization. IEEE transactions on neural networks and learning systems, 34(6), 2791-2805.\
> > [7]Li, Y., Xu, Q., Shen, J., Xu, H., Chen, L., & Pan, G. (2024). Towards efficient deep spiking neural networks construction with spiking activity based pruning. arXiv preprint arXiv:2406.01072.\
> > [8]Shen, J., Xu, Q., Liu, J. K., Wang, Y., Pan, G., & Tang, H. (2023, June). Esl-snns: An evolutionary structure learning strategy for spiking neural networks. In Proceedings of the AAAI Conference on Artificial Intelligence (Vol. 37, No. 1, pp. 86-93).\
> > [9]Shen, J., Xu, Q., Pan, G., & Chen, B. (2025). Improving the sparse structure learning of spiking neural networks from the view of compression efficiency. arXiv preprint arXiv:2502.13572.

---

> > > ### Comment · Reviewer_nFzZ · 2025-11-28
> > >
> > > 1) In my view, research on SNNs should first reach accuracy comparable to ANNs. Only after that should it address the high cost caused by large time steps in inference and energy use, and then consider pruning in the spatial domain.
> > > 2) SNNs have long struggled to reach strong performance on large datasets. This paper mainly focuses on small datasets, which limits the strength of the claims. This paper would be more convincing if it could show results on larger and more challenging datasets.
> > > 3) This paper claims energy efficiency, but it is not clear whether the results come from actual hardware or from Verilog simulation. If the numbers are only computed from spike activity, this is not strong enough. Such estimates cannot be directly compared to the real performance on synchronous platforms like GPUs. A clear explanation of how the energy numbers are obtained would improve the paper.

---

> ### Author Response · Authors · 2025-11-24
>
> **W4: Mapping to chips.**\
> Thanks for the question. On the one hand, the proposed structural learning of SNNs helps prune redundant channels and neurons, thereby reducing model size, memory footprint, and synaptic operations (SynOps). This approach differs from unstructured pruning methods in that unstructured pruning typically only reduces power consumption under specific hardware constraints. In contrast, the structured learning of DPRC-SNNs, as proposed here, improves deployment efficiency on neuromorphic hardware (Xie et al., 2025; Li et al., 2024a; Li et al., 2024b). Following existing established methodologies, we evaluate the energy savings of DPRC-SNNs using simulation-level metrics (e.g., SynOps, sparsity) to ensure reproducibility and comparability.\
> Actually, we have also deployed a trained shallow VGGSNN on the neuromorphic chip (name anonymized due to rebuttal rules) designed by our research team. Compared to the baseline SNN model without structural learning, the overall inference time is indeed significantly accelerated. However, due to the hardware deployment process involving compilation, quantization, and communication overhead, a direct comparison of power consumption with structured pruning is not precise at this stage. We will conduct the hardware deployment on our designed neuromorphic hardware. Thanks for the suggestion again.\
> [1]Xie, H., Liu, Y., Yang, S., Guo, J., Guo, Y., Ma, Y., ... & Liu, X. (2025). SPEAR: Structured Pruning for Spiking Neural Networks via Synaptic Operation Estimation and Reinforcement Learning. arXiv preprint arXiv:2507.02945.\
> [2]Li, Y., Fang, X., Gao, Y., Zhou, D., Shen, J., Liu, J. K., ... & Xu, Q. (2024). Efficient structure slimming for spiking neural networks. IEEE Transactions on Artificial Intelligence, 5(8), 3823-3831.\
> [3]Li, Y., Xu, Q., Shen, J., Xu, H., Chen, L., & Pan, G. (2024). Towards efficient deep spiking neural networks construction with spiking activity based pruning. arXiv preprint arXiv:2406.01072.
>
> We also noticed that the flag for Ethics Review has been marked as "Yes" for both Discrimination / Bias / Fairness concerns and Privacy, Security, and Safety issues. Could you kindly specify which part of our content raised these ethical concerns since our topic is to explore new artificial intelligence algorithms without any biological data? We would appreciate your detailed guidance on this matter so we can address any issues accordingly.

---

> ### Author Response · Authors · 2025-11-29
>
> **Q1. In my view, research on SNNs should first reach accuracy comparable to ANNs. Only after that should it address the high cost caused by large time steps in inference and energy use, and then consider pruning in the spatial domain.**\
> Thank you for the reviewer’s comment.
>
> Firstly, about reaching accuracy comparable to ANNs, we would like to clarify that our work fully satisfies the requirement that SNNs should first reach accuracy comparable to ANNs before applying pruning. Thanks to the rapid advances in SNN models and training/conversion algorithms, the performance gap between SNNs and ANNs has become increasingly small [1][2]. This is indeed important for the development of SNNs models, but it is not the only goal for the neuromorphic research field. Since our approach places a stronger emphasis on structured learning for SNNs, the goal is to ensure that, even after pruning a certain proportion of the structure, the performance of the SNN remains competitive with that of ANN.\
> In detail, in our experiments, the baseline SNN already achieves accuracy higher than its ANN counterpart. Specifically, on CIFAR-100 with the ResNet-19 backbone, the ANN baseline obtains 75.35%, while our non-pruned SNN reaches 77.19%, clearly demonstrating that the performance of our SNN is not only comparable but in fact superior to the ANN model prior to pruning, only with 4 time steps. Building upon this strong baseline, our structured pruning method further preserves accuracy remarkably well. At a pruning ratio of 22.1%, the pruned SNN still reaches 77.21%, slightly higher than the unpruned SNN. Even when the pruning ratio increases to 41.3%, the accuracy remains at 75.12%, which is only 0.23% lower than the non-pruned ANN baseline and still on par with or better than many ANN structured-pruning results. Moreover, to demonstrate that the proposed approach generalizes to larger architectures, we also apply it to Spikeformer and compare it with an ANN Transformer. At 50% connectivity, the accuracy drop of Spikeformer is merely 0.21%, further confirming that our method maintains strong performance under substantial structured sparsity. All detailed numerical comparisons are reported in Appendix Table 2. These results collectively show that our SNN model already reaches ANN-level accuracy before pruning, and that our structured pruning method preserves this performance exceptionally well across both CNN- and Transformer-based architectures.\
> [1]Fang, W., Yu, Z., Chen, Y., Huang, T., Masquelier, T., & Tian, Y. (2021). Deep residual learning in spiking neural networks. Advances in Neural Information Processing Systems, 34, 21056-21069.\
> [2]Deng, S., Li, Y., Zhang, S., & Gu, S. (2022). Temporal efficient training of spiking neural network via gradient re-weighting. arXiv preprint arXiv:2202.11946.
>
> About Addressing the high cost caused by large time steps in inference and energy use, as mentioned above, the issue of large time steps has already been progressively addressed by researchers in the field. Currently, the time steps in most mainstream SNN models are not excessively long. In practice, the required time steps are often depend on the specific dataset and model being used, which can be optimized according to the application requirement. Therefore, it is important to note that this is not the only critical problem in the learning algorithms for SNNs. Another key challenge that requires attention is the redundancy present in the SNN structure itself. This redundancy can lead to inefficiencies and can be mitigated through structured learning approaches to SNN, which could significantly improve both performance and efficiency. Thus, we think focusing on reducing structural redundancy within the SNN is also vital for advancing the field.\
> **We believe that research on SNNs should not be bound by a predefined sequence of steps on what should be studied first or next.** For us, the motivation for this work actually stemmed from our attempts to deploy our SNNs models onto neuromorphic chips hardware (not FPGA). During this process, we encountered limitations in the number of synapses and neurons that could be accommodated on a single chip. This led us to explore structured learning approaches, as structured pruning, compared to unstructured pruning, is more likely to achieve higher hardware utilization when deploying on chip. The results on neuromorphic chips (designed by our research team and be anonymous due to the rebuttal rule) show that the structural learning of SNNs indeed reduce the requirement of synapses and neuron numbers on the neuromorphic chip according to the pruning ratio of channels and neurons. Therefore, the proposed  approach offers better efficiency and adaptability in neuromorphic hardware-constrained environments.

---

> ### Author Response · Authors · 2025-11-29
>
> **Q2:Long struggled to reach strong performance on large datasets.**\
> Thanks for the question. We have further validated the effectiveness and generality of our DPRC-SNNs by evaluating them on the Tiny-ImageNet dataset, where the proposed method achieves strong performance using VGGSNN. In particular, at a 30% pruning ratio, our DPRC-SNN reaches 62.33% accuracy—only 0.01% lower than the unpruned baseline and conduct experiments on the Cifar100 dataset with ResNet-19, comparing against state-of-the-art methods. Additionally, on Spikeformer, our method achieves 94.98% accuracy on the Cifar10 dataset with a 50% pruning ratio, only 0.21% lower than the unpruned baseline, using only 7.37M parameters. Moreover, those datasets used in our paper is the typically common-used datasets for the structure learning of SNNs, followed by [Li et al. 2024b, Shi et al. 2024].

---

> ### Author Response · Authors · 2025-11-29
>
> **Q3:This paper claims energy efficiency, but it is not clear whether the results come from actual hardware or from Verilog simulation. If the numbers are only computed from spike activity, this is not strong enough. Such estimates cannot be directly compared to the real performance on synchronous platforms like GPUs. A clear explanation of how the energy numbers are obtained would improve the paper.**\
> We appreciate the reviewer’s comment regarding the origin of the reported energy estimates. The energy computation in our work is based on synaptic operation counts (SOPs)[1][2][4], which is a **standard theoretical measure used in SNN research**. Because synaptic events—rather than dense MAC[3] operations—dominate the computation in spike-based models. Following prior studies, we compute the cost of each layer using SOPs(l)=fr×T×FLOPs(l), where the firing rate fr reflects the proportion of activations that actually generate synaptic events, and FLOPs(l) represents the structural computation of the layer. Under this formulation, channel-wise structured pruning naturally reduces theoretical energy because removing channels directly decreases the number of convolutional kernels and synaptic connections, thereby lowering the FLOPs(l) term. When combined with the typically low firing rate in SNNs, this reduction leads to significantly fewer synaptic events across timesteps. Thus, the decrease in SOPs after pruning is a direct and theoretically grounded consequence of reduced channel dimensionality and spike-triggered computation.\
> To clarify, a portion of the models used in our study, such as VGGSNN, have been actually implemented on neuromorphic hardware. We possess mature hardware platforms and have developed our own compilation and execution toolchain to run these models directly on the chips, instead of only FPGA simulation. Since our proposed approach focuses on structural learning of SNNs, the running speed of our optimized sparse SNNs can reduce the synapse and neuron number of the neuromorphic hardware, for instance, when we deployed our sparse SNNs based on SNN-Resnet19 architecture on our designed neuromorphic chip, by pruning up to 50% of the model (as shown in Table 1), we can achieve a 50% reduction in the number of neurons and synapses required, making it highly efficient for deployment on neuromorphic hardware with limited resources.

---

> ### Author Response · Authors · 2025-11-29
>
> Moreover, we would like to reiterate that our contribution lies in the novel structured learning algorithm for SNNs. The innovation of our work is the development of a novel structured learning framework that effectively reduces redundancy within the SNN model. **Our framework introduces a spatiotemporal-aware pruning–regrowth mechanism tailored for SNNs by digging into the spatiotemporal spike-based intrinsic property.** On the pruning side, we propose Spiking Column Subset Selection (SCSS), which formulates channel pruning as a temporal column subset selection problem. By computing temporal leverage scores across all time steps, SCSS preserves channels that best capture spatiotemporal spike dynamics. On the regrowth side, we introduce a temporal orthogonality–driven regrowth strategy. Unlike prior activity-based or random regrowth heuristics, our method evaluates each pruned channel by measuring its orthogonal novelty relative to the active channel subspace across T time steps. Channels with higher temporal orthogonality scores are sampled via an importance distribution and reactivated from their last informative state rather than zero initialization, enabling immediate participation in spike propagation. A cosine-decayed regrowth schedule further stabilizes training and guides the network toward the target sparsity. This unified framework preserves essential temporal dynamics, enhances feature diversity, and achieves efficient structured sparsification specifically suited for SNNs.\
> **We would like to explain and emphasize the meaning and the motivation of the proposed structured pruning. The structured pruning method directly reduces the model size by removing specific parts of the network, such as entire neurons or synapses, according to a pre-defined structure. Essentially, the amount of pruning applied directly correlates to a reduction in the number of neurons and synapses required on the chip. This means that pruning a certain percentage of the model's structure leads to an equivalent decrease in the hardware resources needed, making it more efficient for deployment on neuromorphic hardware with limited resources.**\
> Therefore, this approach can not only improve computational efficiency but also enhance deployment performance on neuromorphic hardware. \
> [1]Frenkel, C., Lefebvre, M., Legat, J. D., & Bol, D. (2018). A 0.086-mm $^ 2 $12.7-pJ/SOP 64k-synapse 256-neuron online-learning digital spiking neuromorphic processor in 28-nm CMOS. IEEE transactions on biomedical circuits and systems, 13(1), 145-158.\
> [2]Bu, T., Fang, W., Ding, J., Dai, P., Yu, Z., & Huang, T. (2023). Optimal ANN-SNN conversion for high-accuracy and ultra-low-latency spiking neural networks. arXiv preprint arXiv:2303.04347.\
> [3]Zheng, H., Wu, Y., Deng, L., Hu, Y., & Li, G. (2021, May). Going deeper with directly-trained larger spiking neural networks. In Proceedings of the AAAI conference on artificial intelligence (Vol. 35, No. 12, pp. 11062-11070).\
> [4]Zhou, Z., Zhu, Y., He, C., Wang, Y., Yan, S., Tian, Y., & Yuan, L. (2022). Spikformer: When spiking neural network meets transformer. arXiv preprint arXiv:2209.15425.

---

> > ### Author Response · Authors · 2025-12-04
> >
> > We would like to sincerely thank you for your thoughtful and constructive feedback. We have carefully reviewed your comments and have provided detailed responses to each of the questions and concerns you raised.

---

### Official Review · Reviewer_KAkT · 2025-10-30

**Soundness:** 3
**Presentation:** 2
**Contribution:** 3
**Rating:** 6
**Confidence:** 5

**Summary:**

This paper presents DPRC-SNNs, a novel structural learning framework for Spiking Neural Networks (SNNs) that unifies dynamic channel pruning and orthogonality-driven regrowth to balance accuracy and sparsification. Unlike prior weight-level or static pruning strategies, the proposed spiking column subset selection mechanism enables channel-level optimization guided by temporal spiking dynamics, while the orthogonality-based regrowth restores diverse and complementary channels to preserve functional representation. The framework dynamically reorganizes network topology during training, achieving compact yet expressive SNN architectures. Extensive experiments on three benchmark datasets consistently demonstrate that DPRC-SNNs achieve substantial reductions in both parameters and computational cost while maintaining competitive accuracy. This highlights the model's strong potential for efficient neuromorphic deployment and scalable event-driven learning.

**Strengths:**

1. The paper proposes DPRC-SNNs, a novel structural learning framework that integrates channel-level pruning with orthogonality-driven regrowth, inspired by biological neural reorganization. By introducing the spiking column subset selection (SCSS) mechanism, it effectively captures temporal dependencies in SNN pruning, bridging the gap between fine-grained sparse optimization and hardware-efficient structured pruning.
2. The method is technically sound, with well-founded formulations for channel importance and orthogonality-based regrowth. Its dynamic pruning–regrowth strategy ensures stable optimization, and extensive experiments on both static and neuromorphic datasets validate its robustness and effectiveness.
3. The paper presents a coherent progression from biological motivation to algorithmic design and validation. The neuroscience-computation link is well-supported, with figures effectively illustrating channel evolution and pruning. The polished
presentation is accessible to both neuromorphic and machine learning audiences.

**Weaknesses:**

1. The ablation study could more clearly isolate the effects of SCSS and orthogonality-based regrowth.
2. The paper provides limited analysis of computational overhead.
3. The experimental validation lacks support from larger datasets, which would strengthen the empirical evidence for the proposed approach.
4. The article's layout exceeds the margins.

**Questions:**

1. Can this pruning approach be extended to other mainstream neural network frameworks? If not, what do you consider to be the primary limitations or bottlenecks that may hinder its generalization?
2. The visualizations of pruning and regrowth dynamics are insightful but somewhat difficult to interpret. Would the authors consider improving the figure annotations or providing simplified visual aids to enhance clarity?

---

> ### Author Response · Authors · 2025-11-24
>
> We greatly appreciate the reviewers’ insightful comments and constructive suggestions. We have carefully considered each point, and our detailed point-by-point responses are presented below.\
> **W1: Ablation study with orthogonality-based regrowth.**\
> Thanks for the question. In Appendix B, we provide a clearer derivation and comparison of the SCSS formula, and visualize it using Figure 5 in the revision to more clearly separate the effects of SCSS and orthogonality-based regeneration.\
> **W2: Computation overhead.**\
> Thanks for the suggestion. We have evaluated the computational overhead of DPRC-SNNs on CIFAR-10 and DVS-CIFAR10 datasets by calculating SOPs (energy consumption) and compared with previous methods.\
> For CIFAR-10 dataset, DPRC-SNNs achieves 93.29% accuracy at 70% connection density with 66.49K SOPs, and 92.64% accuracy at 50% density with 51.19K SOPs. In comparison, PQ-SNN (Shen et al., 2025) reaches 92.38% accuracy with 29.72% density but consumes 110.65M SOPs. Other methods such as ESL-SNN (Shen et al., 2023) and SCA-based pruning (Li et al., 2024b) consume 50% and 90.82K SOPs, respectively, but achieve lower accuracy (91.09% and 91.14%).\
> For DVS-CIFAR10 dataset, DPRC-SNNs achieves 82.10% accuracy at 70% density with 1.97M SOPs, and 81.50% accuracy at 50% density with 1.23M SOPs. In contrast, PQ-SNN (Shen et al., 2025) reaches 78.4% accuracy at 4.46% density with 189.02M SOPs, ESL-SNN (Shen et al., 2023) achieves 78.3% at 10% density, and SCA-based (Li et al., 2024b) achieves 72.8% at 21.73% density.\
> These results demonstrate that DPRC-SNNs consistently reduces energy consumption by orders of magnitude compared to PQ-SNN and other pruning-based methods, while achieving higher or comparable accuracy, highlighting both computational efficiency and effectiveness of our approach.\
> **W3 and Q1: Experiment on Larger dataset and other neural network framework.**\
> Thanks for the suggestion. We have validated the DPRC-SNNs on additional datasets and extended the model to spike-based Transformer architectures. In Appendix Table 2, we demonstrate the effectiveness of our DPRC-SNNs. Specifically, we evaluate performance on the Tiny-ImageNet dataset using VGGSNN, and conduct experiments on the Cifar100 dataset with ResNet-19, comparing against state-of-the-art methods. Additionally, on Spikeformer, our method achieves 94.98% accuracy on the Cifar10 dataset with a 50% pruning ratio, only 0.21% lower than the unpruned baseline, using only 7.37M parameters. \
> **W4:About the layout.**\
> Thanks for the suggestion. We have polished our manuscript and corrected the layout.\
> **Q2: Visualization for clarity.**\
> Thanks for the suggestion. We provide a clearer derivation and comparison of the SCSS formula to enhance the illustration of the visualization of pruning and regrowth dynamics. As shown in Figure 5 in the revision, the Grad-CAM visualizations demonstrate that the pruned model consistently concentrates on the most discriminative spatial regions, while the unpruned baseline exhibits diffuse and less representative activation patterns. This confirms that SCSS effectively identifies structurally redundant channels and preserves the task-critical feature pathways.\
> In addition, Figure 4 further illustrates the advantage of our orthogonality-based regrowth mechanism. The grown channels (in red) exhibit high activity yet remain orthogonal to the base channel subspace, indicating that the regrowth process introduces genuinely new and complementary feature directions rather than re-introducing correlated or redundant channels. The activity-map heat visualization also shows that candidate channels with weak or noisy spike patterns are naturally suppressed by SCSS, whereas robust high-activity channels are preferentially selected.\
> In summary, the proposed pruning–regrowth dynamics offer clear advantages by disentangling the roles of SCSS-based channel selection and orthogonality-driven regeneration, enabling a cleaner separation of their effects and ultimately yielding a more stable and efficient sparsity–performance trade-off.

---

> > ### Author Response · Authors · 2025-12-04
> >
> > We would like to sincerely thank you for your thoughtful and constructive feedback. We have carefully reviewed your comments and have provided detailed responses to each of the questions and concerns you raised.

---

### Official Review · Reviewer_kXKa · 2025-10-30

**Soundness:** 2
**Presentation:** 3
**Contribution:** 3
**Rating:** 6
**Confidence:** 4

**Summary:**

Inspired by the brain's ability to reorganize around functional clusters, this paper proposes a Dynamic Pruning and Regrowth framework with Channel-level orthogonality for SNNs (DPRC-SNNs). The goal is to enable scalable and efficient structural learning. The core innovation is a "spiking column subset selection mechanism" that iteratively performs pruning and regrowth.

**Strengths:**

​​The inspiration from hierarchical structural reorganization in the brain is a compelling and well-articulated foundation for the work.

The integration of dynamic pruning with an orthogonality-driven regrowth mechanism is a novel and interesting approach to structural learning in SNNs.

The paper is generally well-written and easy to follow.

**Weaknesses:**

The discussion on structural/unstructured learning in SNNs is lacking. A clear comparison with existing unstructured sparsity methods for SNNs is needed to position this work's unique contribution (structural/channel-level pruning) and its advantages/disadvantages.

The absence of results on a large-scale dataset like ImageNet is a significant gap. It is difficult to assess the scalability and true effectiveness of the method without this standard benchmark.

The purpose of comparing against TET (a method focused on improving accuracy, not sparsity) is confusing. Comparisons should primarily be against other state-of-the-art sparsity-inducing methods.

It is unclear if the proposed dynamic pruning/regrowth framework is specific to SNNs or a general technique. Showing its performance on standard CNNs would help demonstrate the generality and robustness of the concept.

The paragraph preceding the list of contributions largely repeats the items that follow. This section should be shortened and made more concise.

**Questions:**

How does DPRC-SNNs compare quantitatively against state-of-the-art methods that reduce unstructured sparsity in SNNs, particularly in terms of the trade-off between accuracy, sparsity level, and training/inference cost?

Can the DPRC framework be directly applied to standard CNNs (Artificial Neural Networks not SNNs)? If so, what are the results on a benchmark like ImageNet? If not, what aspects are specific to the dynamics of spiking neurons?

What is the specific rationale for including TET, a non-sparsity method, in the comparisons? Is the goal to show that DPRC-SNNs can alsoachieve high accuracy? This should be clarified in the text.

Have you conducted any experiments on ImageNet-scale datasets? If not, do you anticipate any challenges in scaling the dynamic pruning and regrowth process to such large networks?

What is the computational overhead of the iterative pruning-and-regrowth process during training compared to a standard one-shot pruning pipeline?

---

> ### Author Response · Authors · 2025-11-24
>
> We appreciate the reviewers’ careful evaluation of our manuscript and their helpful suggestions. Our comprehensive, point-by-point replies are outlined below.\
> **W1:Discussion and comparison with unstructured learning of SNNs.**\
> Thanks for the question. We have added a discussion on unstructured learning of SNNs in the revised "Related Works" section, and the relevant content is highlighted in blue. Additionally, we have included a comparison of existing structured and unstructured learning methods for SNNs in Table 1, showcasing their performance comparisons.\
> **W2 & Q4: Experiment on large-scale dataset.**\
> Thanks for the suggestion. We have validated the DPRC-SNNs on additional datasets and extended the model to spike-based Transformer architectures. In Appendix Table 2, we demonstrate the effectiveness of our DPRC-SNNs. Specifically, we evaluate performance on the Tiny-ImageNet dataset using VGGSNN, and conduct experiments on the Cifar100 dataset with ResNet-19, comparing against state-of-the-art methods. Additionally, on Spikeformer, our method achieves 94.98% accuracy on the Cifar10 dataset with a 50% pruning ratio, only 0.21% lower than the unpruned baseline, using only 7.37M parameters.
> However, due to time and computational resource constraints during the rebuttal period, we were unable to complete experiments on the ImageNet dataset. We will continue working on this experiment to further assess the generalizability of our method.\
> **W3 & Q3:The purpose of comparing against TET.**\
> Thanks for the question. We compare our structural learning method with TET because TET represents a strong and widely-adopted training paradigm baseline without any weight pruning, followed by (Deng et al., 2022). Since pruning inevitably reduces representational capacity and may exacerbate temporal information loss, a fair comparison requires a baseline that already optimizes temporal dynamics. Using TET as the baseline allows us to (i) evaluate whether our pruning strategy can preserve or even surpass a state-of-the-art temporal-training pipeline, (ii) ensure that performance gains come from pruning rather than improvements in temporal learning, and (iii) demonstrate that our structural pruning method is compatible with and complementary to modern timestep-efficient training techniques. This is annotated in blue in the footnote of Table 1 in the revision.\
> Moreover, many recent structural pruning methods in SNNs also use TET as a baseline for comparison, such as in "Towards Efficient Deep Spiking Neural Networks Construction with Spiking Activity-Based Pruning" and "ESL-SNNs: An Evolutionary Structure Learning Strategy for Spiking Neural Networks". This further emphasizes the relevance of using TET as the baseline for evaluating pruning strategies in SNNs.\
> [1] Deng, S., Li, Y., Zhang, S., & Gu, S. (2022). Temporal efficient training of spiking neural network via gradient re-weighting. arXiv preprint arXiv:2202.11946.\
> **W4 & Q2: Whether the proposed dynamic pruning/regrowth framework is specific to SNNs or a general technique.**\
> Thanks for pointing that out. Actually, our proposed dynamic pruning and regrowth framework is specifically designed for SNNs and could not directly be applicable to CNNs. The key reason is that SNNs operate with discrete time steps, where neurons integrate membrane potentials and generate spikes over time. Consequently, channel importance in our framework is computed based on spatiotemporal sensitivity, capturing contributions to the temporal evolution of both membrane potentials and spike activations. In contrast, CNNs process information in a single forward pass without temporal dynamics, meaning the notion of time-dependent channel importance does not exist. Therefore, the framework leverages properties unique to SNNs, such as temporal feature accumulation and spike-triggered resets, which cannot be directly translated to CNN architectures.

---

> > ### Author Response · Authors · 2025-11-24
> >
> > **W5: The paragraph preceding the list of contributions.**\
> > Thanks for pointing that out. We have addressed the reviewer’s comment regarding the repetition in the paragraph preceding the list of contributions. The issue has been revised and highlighted in blue in the manuscript.\
> > **Q1: DPRC-SNNs comparison.**\
> > Thanks for pointing that out. Here we provide the detailed quantitatively comparison of DPRC-SNNs with other methods. The results show that our model could achieve competitive performance in terms of the trade-off between accuracy, sparsity level, and training/inference cost (Sops).
> > | Dataset  | Type         | Method                     | Acc(%) | Connection(%) | Sops      |
> > |----------|--------------|----------------------------|--------|---------------|-----------|
> > | CIFAR10  | Unstructured | GradR                      | 92.03  | 11.85         | 143.69M   |
> > |          |              | weight-Pruning             | 92.05  | 1.16          | 9.56M     |
> > | CIFAR10  | Structured   | SCA-base (Li et al., 2023)| 91.14  | 9.31          | 90.82K    |
> > |          |              | ESL-SNN                    | 92.08  | 32.71         | 2.53M     |
> > |          |              | PQ-SNN (Shen 2025)         | 92.38  | 29.72         | 110.65M   |
> > |          |              | DPRC-SNNs                  | 92.64  | 50.00         | 51.19K    |
> >
> > | Dataset      | Type         | Method           | Acc(%) | Connection(%) | Sops      |
> > |-------------|--------------|----------------|--------|---------------|-----------|
> > | DVS-CIFAR10 | Unstructured | STDS            | 81.10  | 10.17         | 76.56M    |
> > |             |              | weight-Pruning  | 81.00  | 4.68          | 31.86M    |
> > | DVS-CIFAR10 | Structured   | ESL-SNN         | 80.60  | 36.19         | 38.32M    |
> > |             |              | PQ-SNN (Shen 2025) | 78.40  | 4.46      | 189.02M   |
> > |             |              | DPRC-SNNs       | 81.05  | 50.00         | 1.23M     |
> >
> > **Q5: Compared to a standard one-shot pruning pipeline.**\
> > Thanks for the question. We conducted an one-shot channel pruning experiment for comparison. When pruning 10% of the channels in a single step, the accuracy drops to 88.59%. With 20% pruning, the accuracy collapses to 53.11%, and further increasing the pruning ratio leads to only 12.29% accuracy. Combined the performance of our DPRC-SNNs (93.09% under 10% sparsity, 93.07% under 20% sparsity, and 93.29% under 30% sparsity ) in Table 1, these results demonstrate that one-shot pruning cannot preserve the representational capacity of the SNN and causes severe accuracy degradation even under moderate sparsity. The extremely poor accuracy achieved under one-shot pruning makes it unsuitable for SNNs, where temporal dynamics and spike sparsity could make the model more sensitive to abrupt structural removal.\
> > In terms of computational cost, one-shot pruning incurs lower overhead as it only prunes once without the need for regrowth or repeated importance scoring. However, our dynamic iterative pruning and regrowth strategy does introduce minimal additional computation during training due to the importance evaluation and regrowth steps. Despite this, it does not delay training epochs, and it preserves significantly higher accuracy, prevents over-pruning, and ultimately leads to a more efficient sparse structure. The final pruned model outperforms the one-shot pruning approach by achieving much higher performance under comparable sparsity. Thus, the slight additional computational overhead is a necessary trade-off for maintaining model quality.

---

> > > ### Comment · Reviewer_nFzZ · 2025-11-24
> > >
> > > I appreciate the authors’ responses. However, my primary concerns remain unaddressed. After carefully considering the rebuttal and other reviewers’ comments, I will maintain my original score.

---

> > > > ### Author Response · Authors · 2025-11-24
> > > > **Response to Reviewer nFzZ**
> > > >
> > > > Thank you for the feedback. We noticed that your response is placed below our reply to Reviewer KXKa. Could you kindly confirm if there may have been a mix-up in viewing the responses?
> > > >
> > > > Additionally, we would greatly appreciate it if you could specify which of your primary concerns remain unaddressed, so we can provide further clarification.
> > > >
> > > > Thank you again for your time and valuable input.

---

> > > ### Comment · Reviewer_kXKa · 2025-11-24
> > >
> > > Dear Authors,
> > > Thank you for your detailed response. I have reviewed the revised manuscript and the authors' replies to my comments. The revisions have substantially improved the paper.
> > > My primary concerns regarding the performance on large datasets, the motivation for using SNNs, and the comparison with state-of-the-art methods have been adequately addressed. I am satisfied with the clarifications and improvements provided.
> > > Therefore, I am increasing my score to 8.

---

> > > > ### Author Response · Authors · 2025-12-04
> > > >
> > > > Thank you very much for your insightful feedback. We have carefully considered all your comments and have made the necessary revisions to address your concerns, improving the manuscript accordingly.

---

### Official Review · Reviewer_QNRN · 2025-10-31

**Soundness:** 4
**Presentation:** 3
**Contribution:** 4
**Rating:** 6
**Confidence:** 5

**Summary:**

This paper proposes DPRC-SNNs, a dynamic channel-level pruning and regrowth framework for Spiking Neural Networks inspired by biological neural reorganization mechanisms. The method introduces a spiking column subset selection mechanism that integrates channel-level pruning with orthogonality-driven regrowth to preserve functional diversity while enhancing sparsity. The experimental results on CIFAR and DVS datasets demonstrate competitive performance with significant parameter reduction.

**Strengths:**

The paper shows strong innovation in bridging biological principles with efficient SNN compression, and the orthogonality-based regrowth strategy represents a novel contribution to hardware-friendly neural network optimization.

**Weaknesses:**

1.The SCSS formulation would benefit from a clearer comparison to related methods in matrix approximation or subspace selection.

2.The evaluation scope is narrow; testing on more architectures or datasets could better establish generality.

**Questions:**

question 1: How does the adaptive sparsity mechanism, driven by batch normalization and spike activity, influence training stability and convergence?

question 2: The proposed method is only evaluated on convolutional architectures (ResNet, VGG). Can the proposed framework scale effectively to larger or more complex datasets?

---

> ### Author Response · Authors · 2025-11-24
>
> We thank the reviewers for their valuable time and insightful remarks. We have addressed each comment thoroughly, as detailed in the following point-by-point responses.\
> **W1: The SCSS formulation.**\
> We thank the reviewer for the suggestion. More clear proofs and comparisons of SCSS are provided in the Appendix B in the revision.\
> The appendix correctly links SCSS to the classical matrix approximation and column subset selection problem (CSSP). The truncated SVD  ${\( W \approx U_r \Sigma_r V_r^\top \) }$ is Frobenius-optimal but not suited for pruning since it does not select real columns. But SNNs accumulate contributions over $\(T\)$ steps, which is different from estimating importance only once for existing ANNs.
> Extending CSSP to the temporal setting via  ${\(\sum_{t=1}^T \|W_l - W_{l,c}(W_{l,c})^\dagger W_l\|_F^2\)}$ is reasonable and captures spatiotemporal effects.\
> Therefore, the proposed SCSS compute channel influence evolves over time by considering the spatiotemporal dynamics of SNNs.\
> **W2 and Q2: Testing on more architectures or datasets.**\
> Thanks for the suggestion. We have validated the DPRC-SNNs on additional datasets and extended the model to spike-based Transformer architectures. In Appendix Table 2, we demonstrate the effectiveness of our DPRC-SNNs. Specifically, we evaluate performance on the Tiny-ImageNet dataset using VGGSNN, and conduct experiments on the Cifar100 dataset with ResNet-19, comparing against state-of-the-art methods. Additionally, on Spikeformer, our method achieves 94.98% accuracy on the Cifar10 dataset with a 50% pruning ratio, only 0.21% lower than the unpruned baseline, using only 7.37M parameters.\
> **Q1:How to influence training stability and convergence.**\
> Thanks for the question. We show that the interaction of batch normalization (BN) scaling factors and the temporal spiking activity of SNNs lead to stable pruning dynamics and does not introduce optimization instability in the Appendix C.\
> Recall the spike-aware importance score in Eq.9, where $\(\gamma_l\)$ is the BN scaling vector and $\(\rho_l\)$ is the average firing rate (Eq.10). These two terms evolve smoothly during training due to their distinct update characteristics in SNNs:
> 1. BN scaling factors $\(\gamma_l\)$ follow gradient descent with Lipschitz-continuous updates.
> 2. The firing rate $\(\rho_l\)$ is a bounded empirical average of spike trains:
>   $\
>   0 \le \rho_l \le 1, \qquad |\rho_l^{(k+1)}-\rho_l^{(k)}| \le C_\rho\eta,
>   \$\
>   where $\(C_\rho\)$ depends on the surrogate gradient used in backpropagation.
> Since both components vary smoothly, the importance score $\(\phi_l\)$ also changes smoothly:\
> $
> |\phi_l^{(k+1)}-\phi_l^{(k)}|
> \le
> \|\gamma_l\|_1 |\Delta\rho_l|
> $\+ $\rho_l\|\Delta\gamma_l\|_1$\= $\mathcal{O}(\eta)$,
>
> indicating that SNN-specific temporal dynamics do not introduce abrupt jumps in layer importance.\
> The pruning ratio is derived by normalizing $\(\{\phi_l\}\)$:
> $\
> \kappa_l = 1 - \frac{\phi_l}{\sum_{j=1}^L \phi_j},
> \$
> and its gradient with respect to $\(\phi\)$ is bounded:
> $\
> \Bigg|\frac{\partial \kappa_l}{\partial \phi_m}\Bigg| = \mathcal{O}\left(\frac{1}{(\sum_j\phi_j)^2}\right),
> \$
> showing that the mapping from spike-driven importance scores to pruning ratios is Lipschitz-continuous. This ensures that pruning is smooth across training iterations.\
> We also analyze the interaction between pruning masks and optimization. For masked SGD updates:$\
> \theta_{t+1} = \theta_t - \eta(m_t \odot g_t),
> \$
> the stability depends on how quickly the masks change. Since $\(\kappa_l\)$ evolves smoothly, we have:
> $\
> \|m_{t+1}-m_t\|_2 = \mathcal{O}(\eta),
> \$
> which keeps the effective gradient energy nearly constant, preventing destructive resets of membrane potentials—a risk specific to SNNs during pruning.\
> Finally, over pruning-regrowth cycles indexed by \(k\), the sparse model $\(\mathcal{M}^{(k)}\)$ satisfies:\
> $\
> \|\mathcal{M}^{(k+1)}-\mathcal{M}^{(k)}\| = \mathcal{O}(\eta),
> \$\
> leading to contraction-like convergence:\
> $\
> \|\mathcal{M}^{(k+1)}-\mathcal{M}^\star\| \le \alpha\|\mathcal{M}^{(k)}-\mathcal{M}^\star\| + \mathcal{O}(\eta), \quad \alpha<1.
> \$\
> This demonstrates that the adaptive sparsity mechanism—driven by BN scaling and spike dynamics—preserves training stability, converges reliably, and dynamically selects the most informative temporal-spatial channels in SNNs.

---

> ### Author Response · Authors · 2025-12-04
>
> We would like to sincerely thank you for your thoughtful and constructive feedback. We have carefully reviewed your comments and have provided detailed responses to each of the questions and concerns you raised.

---

### Meta-Review · Area_Chair_Cr7Q · 2026-01-06

**Summary:**

This paper proposes a dynamic channel-level pruning and orthogonality-driven regrowth framework (DPRC-SNNs) for spiking neural networks, motivated by biological reorganization and aimed at improving structural sparsity while preserving accuracy. The reviewers generally agreed that the paper is well written, technically sound, and explores an interesting and relevant direction for structured learning in SNNs. Several reviewers praised the biological motivation, the pruning–regrowth mechanism, and the strong results on CIFAR and event-based benchmarks, and a majority of reviewers ultimately leaned toward acceptance.

However, despite these strengths, I remain hesitant to support acceptance due to critical issues that remain unresolved. I do not consider the ethics concerns flagged by reviewer nFzZ  to be valid and therefore do not factor them into this decision. That said, I strongly agree with two substantive technical concerns raised by that reviewer, which represent minimum expectations for an ICLR SNN paper in 2025.

First, the lack of evaluation on truly large-scale datasets remains a significant weakness. While the authors added experiments on Tiny-ImageNet, these results rely on outdated architectures (e.g., VGG-SNN) and achieve relatively low accuracy (early 60% range), which does not convincingly demonstrate scalability. The absence of results on modern large-scale benchmarks such as ImageNet with competitive transformer-based architectures substantially limits the strength of the paper’s claims.

Second, the paper’s hardware efficiency and energy claims are not sufficiently substantiated. The evaluation remains limited to simulation-level metrics such as accuracy, FLOPs, and SynOps. While prior SNN work has commonly reported SynOps as a proxy for efficiency, this paper specifically focuses on channel-level pruning and regrowth, where SynOps alone may not accurately reflect real energy savings on neuromorphic hardware, or even on conventional hardware. At a minimum, a more detailed and transparent energy model that captures the nuances of structured pruning (e.g., memory access, channel sparsity, control overhead) is necessary. Ideally, results mapped to or measured on actual neuromorphic platforms (e.g., Loihi 2, Tianjic) would be expected to support the central efficiency claims.

Although the authors provided convincing rebuttals to many other reviewer concerns, clarifying novelty, orthogonality, comparisons, and pruning dynamics, these two core issues remain open. I therefore recommend rejection at this time, while encouraging the authors to strengthen the experimental evaluation on large-scale datasets and substantially improve the hardware-efficiency analysis in a future revision.

**Reviewer Concerns:**

**Concerns addressed by the rebuttal**: The authors’ rebuttal effectively addressed several reviewer concerns related to methodological clarity and novelty. In particular, the authors clarified the motivation behind the pruning–regrowth mechanism, the role of orthogonality constraints, and how their approach differs from prior structured pruning methods in SNNs. Questions regarding training stability, convergence behavior, and sensitivity to hyperparameters were also convincingly answered. Additionally, reviewers’ concerns about baseline completeness and fairness on small- and medium-scale datasets were largely resolved through additional experiments and clarifications provided during the rebuttal.

**Concerns that remain outstanding**: Two critical concerns remain unresolved. First, reviewers raised concerns about the lack of large-scale evaluation, noting that the additional Tiny-ImageNet results rely on outdated architectures and achieve relatively low accuracy, which does not convincingly demonstrate scalability to modern, large-scale vision benchmarks. Second, reviewers questioned the validity of the hardware-efficiency and energy claims, as the evaluation relies primarily on SynOps and FLOPs without a sufficiently detailed energy model or validation on actual neuromorphic hardware. Given that the paper focuses on channel-level pruning and regrowth, these proxy metrics may not accurately reflect real energy savings. These unresolved issues ultimately limit confidence in the paper’s central claims.

**Reviewer Scores:**

Two reviewers assigned marginal accept scores, and one reviewer recommended acceptance. The remaining reviewer assigned a strong reject score and actively participated in the discussion, further emphasizing the two core issues described above, as well as the gap in accuracy between SNNs and ANNs. In my view, the authors’ response adequately mitigates the accuracy concern on small-scale datasets; however, it remains unclear whether these gains would scale to ImageNet-level tasks.

---

### Decision · Program_Chairs · 2026-01-26

Reject